# Improving the Detection and Understanding of Infectious Human Norovirus in Food and Water Matrices: A Review of Methods and Emerging Models

**DOI:** 10.3390/v16050776

**Published:** 2024-05-14

**Authors:** Sahaana Chandran, Kristen E. Gibson

**Affiliations:** Department of Food Science, Center for Food Safety, University of Arkansas System Division of Agriculture, Fayetteville, AR 72704, USA; sc112@uark.edu

**Keywords:** human norovirus, human intestinal enteroids, zebrafish

## Abstract

Human norovirus (HuNoV) is a leading global cause of viral gastroenteritis, contributing to numerous outbreaks and illnesses annually. However, conventional cell culture systems cannot support the cultivation of infectious HuNoV, making its detection and study in food and water matrices particularly challenging. Recent advancements in HuNoV research, including the emergence of models such as human intestinal enteroids (HIEs) and zebrafish larvae/embryo, have significantly enhanced our understanding of HuNoV pathogenesis. This review provides an overview of current methods employed for HuNoV detection in food and water, along with their associated limitations. Furthermore, it explores the potential applications of the HIE and zebrafish larvae/embryo models in detecting infectious HuNoV within food and water matrices. Finally, this review also highlights the need for further optimization and exploration of these models and detection methods to improve our understanding of HuNoV and its presence in different matrices, ultimately contributing to improved intervention strategies and public health outcomes.

## 1. Introduction

### 1.1. Human Norovirus

Human norovirus (HuNoV) is the main causative agent for acute gastroenteritis worldwide [1,2,3]. This pathogen is highly contagious and causes self-limiting infections in healthy individuals but severe complications in immunocompromised individuals [4,5]. HuNoVs are constantly evolving, and as there is only limited cross-protection across strains, individuals can become infected with multiple strains of HuNoV throughout their lifetime [5,6]. These characteristics have made HuNoV one of the most effective disease-causing human pathogens among all age groups. According to the World Health Organization’s assessment of all reviewed cases of foodborne illness, HuNoV was the leading cause of both foodborne illness and foodborne deaths [7]. In the United States, HuNoV is responsible for an estimated 21 million gastroenteritis cases yearly, with more than 70,000 hospitalizations and 800 deaths [8]. The noroSTAT data showed 759 norovirus outbreaks between August 2023 and February 2024 reported by 15 noroSTAT-participating US states [9].

Noroviruses (NoVs) belong to the family of Caliciviridae and are single-stranded RNA non-enveloped viruses [10,11]. Based on the protein diversity of the major viral capsid protein (VP1), NoVs are classified into ten genogroups (GI-GX), which are further divided into 49 genotypes. Division into genotypes is based on the complete VP1 amino acids sequence with a 15% sequence difference as a cut-off threshold for new genotypes [10]. Among these genogroups, GI, GII, and GIV viruses are the ones most associated with human infections, with genogroups GVIII and GIX becoming associated with human infections only recently [6,10,12]. While most of the NoVs in these genogroups circulate among the human population with variable incidences, GII primarily governs the transmission of HuNoV globally, particularly the GII.4 variants due to their intra-genotype recombination [2,3,13]. The variant GII.4 2012 Sydney has been reported to be the most predominant strain, causing up to 85% of the global HuNoV epidemics between the years 2000 to 2013 [4,12]. However, in the winter of 2014/15 a novel GII.17[P17] strain was found to be the dominant strain in some parts of Asia [13]. Non-GII.4 viruses such as GII.2[P16] and GII.3[P16] have also been observed recently, with increased incidence causing large outbreaks in multiple countries [14]. Between 1 September 2023 and 31 March 2024, CaliciNet—the national norovirus outbreak surveillance network in the United States—recorded 125 norovirus outbreaks nationwide. Within this timeframe, the GI.5[P5] genotype was found to be accountable for up to 14.4% of these outbreaks [15]. Additionally, other significant genotypes contributing to over 10% of the outbreaks during this period include GII.17[P17], GII.4 Sydney[P16], and GII.4 untypeable[P16] [15].

### 1.2. Transmission Routes

HuNoV transmission occurs via fecal–oral and vomit–oral pathways by four general routes: direct person-to-person, foodborne, waterborne, and environmental fomites. The most common route is person-to-person, which has been the cause of most outbreaks and sporadic diseases. In Japan, a study was carried out to identify the most common transmission route in adults between 2015 and 2019. Person-to-person transmission accounted for approximately 30% of cases, while foodborne transmission was associated with 20% [16]. As only 2800 genomic equivalents are required to cause an infection [17], the virus spreads easily through aerosolized droplets and environmental contamination [18]. Once in the environment, HuNoV can withstand freezing temperatures and temperatures up to 60 °C, is often resistant to common sanitizers and disinfectants (e.g., chlorine bleach at low concentrations, quaternary ammonium compounds, ethyl alcohol), can persist on surfaces for at least two weeks, and can remain detectable in groundwater for over 3 years, with infectivity lasting at least 61 days [5,19,20]. The second most common transmission route is through contaminated food, where infected food handlers typically transfer the virus to food products [4,19,20]. If HuNoV-contaminated water is used to irrigate crops, the virus may be present in the fruits and vegetables that are consumed raw. In addition, shellfish grown in water contaminated with human sewage may also contain HuNoV because shellfish are filter-feeders [21]. Their feeding process involves drawing water into their gills, where suspended food particles are captured and directed into the digestive tract. Since HuNoVs present in environmental water can adhere to these particles, they can easily infiltrate shellfish during their filtering mechanism. Once inside the digestive tract of bivalve mollusks, HuNoV attaches to HBGA-like carbohydrates in the gastrointestinal epithelial cells, proving to be a challenge to eliminate [22].

Due to the globalization of the food industry, the foodborne transmission of HuNoV paves the way for the spread of different genotypes to various parts of the world, causing outbreaks of multiple HuNoV strains at the same time, often leading to viral recombination events [4]. A recombinant norovirus is characterized by its clustering with two separate norovirus genotypes when subjected to phylogenetic analysis of two distinct regions, typically the capsid [VP1] and the RNA-dependent RNA polymerase [P], within the genome [23]. Numerous recombinant viruses have been documented in the literature, yet only a select few are considered epidemiologically significant [24,25,26]. Among these are GII.4 Sydney 2012[P31], GII.4 Sydney 2012[P16], GII.2[P16], GII.6[P7], and GII.3[P12] [26,27,28,29,30,31,32].

## 2. Foodborne Transmission of Human Norovirus

### 2.1. Human Norovirus Contamination of Fresh Produce

Fresh-produce-related infections caused by HuNoV are a widespread global concern, posing a significant threat to public health [33]. Various farming activities, such as the use of contaminated water for irrigation, the application of organic fertilizers (e.g., manure and compost), on-farm hygiene, handling, packaging, and processing, represent critical points where fresh produce can become contaminated by HuNoV [34]. In the United States, HuNoV has been identified as the causative agent in 59% of fresh-produce-related outbreaks, and in the European Union, it has been associated with 53% of such cases [35]. Norovirus outbreaks have been linked to contaminated leafy greens [36,37,38,39] and soft red fruits such as raspberries and strawberries [40,41].

Ekundayo and Ijabadeniyi (2022) [42] performed a systematic review of the literature related to HuNoV contamination in fresh produce. Later, a meta-analysis of 22 articles was performed to determine the prevalence of HuNoV in fresh produce globally. The results showed an overall prevalence of HuNoV in fresh produce at 9.3%. The overall prevalence of genogroups GI and GII in fresh produce was found to be 5.3% and 1.7%, respectively, with no significant difference between fruits and vegetables, indicating an equal likelihood of contracting a HuNoV infection from both fruits and vegetables. Furthermore, the detection of both GI and GII HuNoV strains in fresh produce was 0.3%, with no notable difference between fruits and vegetables [42]. It is worth noting that the prevalence of HuNoV GI was higher than that of HuNoV GII in fresh produce. This discrepancy suggests that there may be differences in the environmental survival and stability of GI compared to GII within fresh produce [43,44,45]. Other researchers have also reported that water- and food-borne outbreaks are predominantly caused by HuNoV GI [4,46,47].

### 2.2. Human Norovirus Contamination of Bivalve Shellfish

Outbreaks of acute gastroenteritis linked to the consumption of raw or partially cooked bivalve mollusks, such as oysters, mussels, cockles, and clams, have frequently been reported on a global scale [48,49]. From 2003 to 2017, a total of 51 HuNoV outbreaks occurred worldwide, and a significant majority (61%) were associated with bivalve shellfish [50]. In the United States, three multistate outbreaks associated with NoV-contaminated raw oysters were reported between 2019 and 2023 [51]. Both symptomatic and asymptomatic individuals infected with HuNoV GI and GII shed the virus in large quantities in their feces for extended periods [52]. Raw or inadequately treated sewage can introduce contamination into coastal waters. Bivalve mollusks, owing to their filter-feeding behavior, accumulate pathogens to levels significantly higher than those found in the surrounding water. Notably, in eight HuNoV outbreak investigations conducted between 2009 and 2014 in the United States, HuNoV was consistently detected in all bivalve shellfish samples implicated in HuNoV outbreaks [53]. Furthermore, a 100% genetic match was found between the shellfish and the clinical strains associated with each of the outbreaks [53]. The degree of HuNoV contamination in shellfish is strongly influenced by various factors, including the species of shellfish and environmental conditions including season, rainfall, temperature, water quality, and tidal changes [54,55]. Interestingly, the detection rate of HuNoV varies among different shellfish species within the same area. Mussels, for instance, exhibit a higher prevalence of positive tests for HuNoV GII compared to oysters, clams, and cockles. Conversely, the detection rate of HuNoV GI is higher in cockles than in mussels and clams [56]. Additionally, research indicates that oysters selectively concentrate GI strains over GII strains by binding to an A-like carbohydrate structure, similar to HBGA, that is present in the digestive ducts [57]. Hence, while GII viruses require 1200 RNA copies/L to bioaccumulate 1 viral RNA copy/g of oyster tissue, GI viruses require only 30 RNA copies/L [57]. These results show the distinct prevalence patterns of HuNoV among different shellfish.

### 2.3. Human Norovirus Contamination of Water Sources

Human noroviruses have been detected in various water sources, including surface water (i.e., lakes and rivers), groundwater, tap water, seawater, marine water, and wastewater, as reported in multiple studies [58,59,60,61,62,63,64]. In raw sewage, HuNoV has been measured at levels ranging from 3.8 to 6.66 log copies/L for HuNoV GI and 3.8 to 7.3 log copies/L for HuNoV GII [65]. Although sewage treatment processes can reduce HuNoV levels, the high resistance of HuNoV can lead to its persistence in effluent water even after treatment [66]. A comprehensive meta-analysis encompassing 61 published studies on HuNoV contamination in water sources revealed varying degrees of prevalence. River water exhibited the highest estimated prevalence, at 43.5%, followed by estuarine water, composite water (river–estuarine water, river–groundwater, river–lake water, and groundwater–brackish water), marine water, groundwater, and lake water, with reported prevalences of 30.6%, 27.9%, 25.9%, 19.7%, and 2.2%, respectively. Additionally, the prevalences of GI, GII, and both GI and GII genogroups in natural water sources were reported at 16.4%, 20.6%, and 12.8%, respectively [67]. In a study assessing the occurrence of human enteric viruses in Wisconsin household wells over a year, HuNoV was detected in one out of 50 wells at a single sampling time. This sampling occurred four times throughout the study period [68]. Another study aimed to investigate the number of acute gastrointestinal illnesses resulting from the waterborne transmission of enteric viruses in tap water across a 12-week duration [69]. Here, the authors revealed that at a HuNoV mean concentration of one genomic copy per liter, the incidence ratio of gastrointestinal illness increased by 30% [69]. Collectively, these results show the potential risk of HuNoV contamination of water sources.

## 3. Detection of Human Norovirus in Food and Water

### 3.1. Extraction and Concentration Methods

As HuNoVs cannot be readily cultured in a cell culture system, in order to detect the presence of HuNoV in food and water matrices, researchers have heavily relied on molecular methods such as RT-qPCR, where the HuNoV genomic fragment is amplified, and eventually, the amount of viral RNA present is quantified. However, the initial extraction (or elution) of viruses from each matrix differs depending on the components of the matrix. The most recent standard method (ISO 15216-2:2019) was published in 2019 for detecting HuNoV genogroups GI and GII from food matrices including soft fruits, leaf, stem and bulb vegetables, bottled water, and bivalve molluscan shellfish [70]. The protocol involves eluting the viruses from the vegetable and fruit matrices using an elution buffer followed by agitation and a concentration step using polyethylene glycol (PEG) precipitation. For extracting viruses present in water, a concentration step using adsorption–elution on an electropositive microporous filter is performed, followed by ultrafiltration to concentrate the virus present. To extract the virus from bivalve shellfish, the digestive glands are dissected from the animal, followed by homogenization and treatment with proteinase K. Viral RNA is then isolated from all the matrices using the chaotropic agent guanidine thiocyanate to disrupt the viral capsid. RNA is then adsorbed to a silica column to assist purification through several washing stages, followed by the elution of RNA from the column.

El-Senousy and co-authors (2013) [44] validated the ISO method for HuNoV detection in 720 naturally contaminated samples of fresh produce (green onion, watercress, radish, leek, and lettuce) and in 144 irrigation water samples. PEG precipitation and organic flocculation (OF) were used as virus-concentration procedures. Results showed that the virus prevalence of fresh produce with PEG precipitation was significantly higher when compared to OF, ranging from 28.0% to 48.0% and from 14.0% to 18.8%, respectively. In irrigation water, HuNoV was present in 31.9% of water samples with PEG precipitation and 25% of water samples with OF. In another study, four different virus extraction/concentration methods, namely ultrafiltration, immunomagnetic separation, ultracentrifugation, and PEG precipitation, were compared for recovering GII viruses present on artificially inoculated lettuce, sliced ham, and raspberries [71]. The results showed that the PEG precipitation method had the most reproducible results across all food matrices, although ultracentrifugation yielded the highest recovery rate in lettuce and ham. Similar results were observed for the recovery of norovirus from frozen strawberries as well, where PEG precipitation yielded a better detection rate compared to ultrafiltration [72,73] and other virus extraction methods such as direct lysis, porcine gastric mucin-coated magnetic beads, and TRI Reagentä from frozen strawberry [73]. To concentrate HuNoV from 100 g of lettuce samples, a secondary concentration step using ultrafiltration after PEG precipitation has been shown to have a recovery percentage of 11.4% [74]. Kim et al. (2008) [75] optimized the process of extracting norovirus from the surfaces of grapes, strawberries, and frozen raspberries. Six different buffers were tested to elute norovirus from fruit surfaces and confirmed that 3% beef extract was the most effective. Additionally, PEG molecular weight, along with incubation temperature and duration, were assessed for the concentration of norovirus in samples. The results showed that PEG10,000, used for 4 h at room temperature, efficiently concentrated norovirus. Further, five RNA extraction methods were evaluated, including heat-release, QIAampâ Viral RNA Mini Kit, magnetic beads, TRIzolâ reagent, and immunomagnetic separation with magnetic Dynabeadsä, and the QIAampâ Viral RNA Mini Kit was reported as the most efficient [75]. Nevertheless, given that this study was conducted 15 years ago, it is imperative to reassess newer kits available on the market.

Tunyakittaveeward et al. (2019) [76] compared the ISO 15216 extraction with an adsorption–elution method to investigate the presence of HuNoV in oysters purchased from a local market in Bangkok, Thailand. Seventy oyster samples were analyzed, and the results showed that with the ISO (proteinase K treatment) method, HuNoV genogroup GI was detected more, whereas with the adsorption–elution method, genogroup GII was detected more [76]. With PEG precipitation, the recovery rate of HuNoV from oysters was found to be 7-fold higher than that of the ISO 15216 method [77]. Furthermore, treatment with cetyl trimethyl ammonium bromide after RNA extraction and lithium chloride precipitation has been shown to provide RNA with high extraction efficiency from mollusks (the recovery rate is 4.3 times higher than that of ISO 15216) and a negligible inhibitory effect on RT-qPCR [78]. In another study, blue mussels were bioaccumulated with NoV GI and GII strains. The extraction of viruses from the shellfish tissue revealed distinct characteristics in terms of elution for the two virus strains. Proteinase K digestion was found to be the preferred method for mussel processing, particularly when screening for a broad range of HuNoV strains is required. However, this method had a slightly lower sensitivity in detecting the HuNoV GII strain compared to ultracentrifugation [79].

The ISO 15216 method does not include a method to detect HuNoV present in dairy foods, as milk products are not often reported as the source of HuNoV outbreaks. However, when contaminated fruits or other products are added to dairy products, it can lead to an outbreak. Hennechart-Collette and co-authors (2023) [80] showed that the proteinase K-based extraction method could be applied to dairy products such as cheese, milk, yogurt, and dessert cream that were artificially contaminated with HuNoV GI.3 and GII.4. The ISO 15216 method has also been shown to be efficient in extracting and detecting norovirus present in multicomponent foodstuffs where the food product is first eluted in an elution buffer, followed by PEG precipitation [81].

### 3.2. Detection Methods

The detection of norovirus extracted from food and water samples is normally undertaken using nucleic acid amplification techniques such as PCR as it provides high specificity. The gold standard for norovirus quantification currently is quantitative RT-PCR (RT-qPCR), due to its sensitivity and specificity [82]. However, this method requires the construction of a standard curve which might cause inter-laboratory variations. In recent years, droplet digital PCR (ddPCR) has been used to quantify HuNoV present in fresh produce [83,84,85], shellfish [86,87,88,89], and water samples [90]. The results have shown ddPCR to be less sensitive to inhibitory substances from the sample matrix [84], and as there is no need for a standard curve, the inter-laboratory variations would be reduced, as shown in a recent study where nine different labs tested oyster samples spiked with a known amount of norovirus GI and GII [86]. The other alternative method that is widely used for the detection of norovirus from stool samples (i.e., due to their simplicity and faster timeline) is an immunoassay, such as enzyme-linked immunosorbent assay (ELISA). However, these methods have been mostly validated to detect HuNoV present in fecal and blood samples but not when present in food and water samples. Tian and Mandrel (2006) [91] described a method called real-time immune polymerase chain reaction (rtI-PCR) to detect HuNoV capsid proteins in food and fecal samples directly, eliminating the need for virus concentration and purification. In this approach, viral antigens were employed to capture recombinant Norwalk virus-like particles. Despite its assay sensitivity being 10 times higher than that of RT-qPCR, a significant variability in results was observed both within and between experiments. This variability may explain why this method has not been widely adopted for the detection of HuNoV in food samples. Vinje (2015) [82] extensively reviewed the different detection methods employed for identifying norovirus, while Gyawali and colleagues (2019) [92] provided a detailed review specifically focusing on HuNoV detection methods for contaminated shellfish. Recently, a novel method for detecting norovirus in water samples involving a custom-built smartphone-based fluorescence microscope and a paper microfluidic chip was described and was capable of detecting 1 genome copy/μL in deionized water and 10 genome copies/μL in undiluted reclaimed wastewater [93]. As the limit of detection is extremely low in this method, this could be adapted for detecting noroviruses present in food, as well.

### 3.3. Challenges Encountered in Human Norovirus Recovery and Detection

Numerous studies have highlighted the limitation of relying solely on the detection of viral RNA, as it tends to overestimate the presence of infectious virus particles within a sample. In a human volunteer study, researchers explored the correlation between molecular detection results of HuNoV in berries and their implications for public health risks [94]. Twenty participants were tasked with consuming berries they had personally purchased and then submitting product aliquots for RT-qPCR analysis for the presence of HuNoV. Despite none of the twenty participants reporting any symptoms resembling a HuNoV infection after six separate consumption instances, 28 samples were identified as positive for HuNoV GI and/or GII [94]. In another study, a comprehensive analysis was conducted on various types of fresh produce, including 867 samples of leafy greens, 180 samples of fresh soft red fruits, and 57 samples of other fresh produce varieties like tomatoes, cucumbers, and fruit salads [95]. This investigation revealed the frequent detection of HuNoV genomes in fresh produce through RT-qPCR. However, it is important to note that sequence confirmation was often unsuccessful for most of the HuNoV-positive samples. Furthermore, infections or outbreaks were rarely, if ever, found to be linked to the HuNoV-positive samples. Therefore, it is essential to exercise caution when interpreting molecular detection signals in the context of assessing public health risks. Factors within the food matrix, such as PCR inhibitors, can impact the accuracy of estimating the viral RNA’s presence. Notably, substances like carbohydrates and lipids, present in substantial quantities in certain food products, can hinder the sensitivity of RT-qPCR [96,97]. PCR inhibitors present in shellfish, berries, plants, and water samples, and their mechanisms of action in inhibiting PCR reaction, are reviewed in detail by Schrader and co-authors (2012) [98]. Moreover, the quantity of virus present in the food also significantly influences its detectability. For instance, in a study involving ready-to-eat penne salad samples, it was observed that a higher level of GI and GII NoV inocula (approximately 10^6^ NoV genomic copies per 10 g) could be consistently recovered in at least four out of six PCRs [99]. In contrast, lower levels of GI and GII NoV inocula (around 10^4^ NoV genomic copies per 10 g) resulted in recovery in a maximum of three out of six PCRs [99]. This highlights the impact of the HuNoV concentration on its successful recovery. A similar trend was noted in frozen raspberry crumb samples and strawberry puree [100].

Noroviruses remain stable in water sources for long periods of time; therefore, studies are conducted in water bodies to aid in risk assessments. However, routine microbial analyses of water typically involve collecting only around 100 mL samples, which may not allow for the detection of low levels of virus. Additionally, the detection methods employed may have a similarly low detection limit or sensitivity. Therefore, water samples are usually concentrated before testing for the presence of norovirus, during which inhibitors such as bacterial debris, complex polysaccharides, metal ions, and nucleases can also be co-concentrated and potentially inhibit the quantification of target nucleic acids through RT-qPCR [101]. Water samples (*n* = 3193) collected from various sources (surface, groundwater, drinking water, agricultural runoff, sewage) and spiked with hepatitis G virus were evaluated to monitor inhibition during RT-qPCR. The percentage of samples that might have been recorded as false negatives, had inhibition not been addressed, ranged from 0.3% to 71% [101]. In a separate study examining the presence of enteric viruses in tap water, 94 out of 1204 samples analyzed required mitigation for inhibition [69]. Similarly, an inhibitory effect was noted during the analysis of enteric viruses in groundwater, affecting 8% of the samples analyzed [68].

## 4. Cell Culture and Animal Models Available for Detecting Infectious Human Norovirus

Human norovirus has long posed a challenge for in vitro cultivation due to its enteric nature and active replication in the enteroendocrine cells of the intestinal epithelium [102]. Extensive efforts have been made to establish in vitro cultivation systems for HuNoVs in epithelial cells, but these endeavors had proven largely unsuccessful [103], until the development of a 3-dimensional (3D) intestinal epithelial culture [104]. The authors reported the successful replication of both GI and GII viruses in the INT-407 embryonic intestinal epithelial cell line. However, subsequent attempts by various other research groups to replicate this work reported that the 3D cell culture models using INT-407 did not facilitate norovirus replication [105,106,107]. Thus, presently, there are two primary culture systems for HuNoVs: one involves the use of a transformed B-cell line (BJAB cells), and the other employs the human intestinal enteroids (HIEs) system.

BJAB cells have demonstrated the capacity to support the replication of GII.4 Sydney in the presence of HBGA-expressing bacteria or free HBGAs [108]. However, the replication levels achieved in these cells have not been sufficient to produce a virus stock, and these results have not consistently been corroborated by other research groups [108,109]. Meanwhile, the HIEs are generated from stem cells isolated from intestinal crypts in human intestinal tissues and have proven successful in cultivating various HuNoV GII genotypes using filtered stool samples [110,111]. Importantly, HuNoV replication in HIEs mirrors epidemiological differences in host susceptibility, which are predicated on genetic variations in the expression of HBGAs tied to a donor’s secretor status [111]. Beyond their utility in studying viral replication and pathophysiology, the HIE cultivation system serves as a valuable platform for evaluating antiviral candidates, conducting neutralization studies, investigating virus inactivation methods, and assessing virus presence in diverse environments such as water, fomites, and shellfish [110,111,112,113,114,115,116,117,118].

A more recent development in HuNoV pathogenesis research is the utilization of the zebrafish larvae model [119]. This model has demonstrated the replication of several strains belonging to the GII genogroup, along with a single strain from the GI genogroup (GI.7), by microinjecting filtered stool samples into the yolk of 3-days-post-fertilization (dpf) zebrafish larvae. The zebrafish larvae model has also been used in evaluating the efficacy of small molecule inhibitors, elucidating the role of HBGAs and microbiota in HuNoV infection, and assessing disinfection treatments for HuNoV inactivation [119,120,121,122]. More recently, a zebrafish embryo model with enhanced efficiency and robustness, compared to the larvae model, has been reported. In this model, HuNoV is injected into the yolk of fertilized zebrafish embryos [123]. The development of culture systems for HuNoV has opened new avenues for research, enabling a deeper understanding of HuNoV replication, pathogenesis, and potential control measures.

## 5. Human Intestinal Enteroid Model for Detecting Infectious Human Norovirus

Human intestinal enteroids, or “mini guts”, are a 3D culture system developed from stem cells isolated from human intestinal tissues [110]. These HIEs contain multiple intestinal epithelial cell types such as enterocyte, goblet, enteroendocrine, and Paneth cells. Whether grown in 3D or as a monolayer, the non-transformed differentiated HIEs recapitulate the human intestinal epithelium and support HuNoV replication [110]. Multiple HuNoV genogroups have been successfully cultivated in HIEs and are listed in Table 1. GII.4 variants that cause the worldwide pandemic of acute gastroenteritis infect HIEs with high efficiency, whereas the addition of human bile was required for the replication of GI.1, GII.3, and GII.17 viruses [110,113].

Costantini and coauthors (2018) [113] tried infecting HIEs with different strains of HuNoV (12 GI, 65 GII, and 3 GIV), and the successful replication of six different strains belonging to the GII genogroup was observed, which included GII.1, GII.2, GII.3, GII.4, GII.14, and GII.17. The success rate with a moderate viral RNA titer (10^3^–10^4^ viral RNA copies/μL) was relatively low. However, several high-viral-load (10^6^ viral RNA copies/μL) GII samples also failed to replicate in HIEs, proving that other currently unknown factors play a role in the replication of HuNoV. It must also be noted that 81% of the fecal samples used in this study were obtained from children < 2 years of age [113]. Therefore, future experiments should aim to use fecal samples collected from adults to see if there is a difference in the replication seen in HIEs, as the fecal samples may contain different host factors such as the composition of the bile and microbiome depending upon the age of the patient. Studies have shown that both bile and the gut microbiome play an important role in HuNoV pathogenesis, and hence it is imperative to study the use of fecal samples collected from individuals belonging to different age groups as well as geographical regions [110,124,125,126,127]. Also, the replication of GI.1 reported previously by Ettayebi et al. (2016) [110] was not able to be reproduced by Costantini and co-authors (2018) [113], as they observed no replication of GI.1 in HIEs even under the same conditions as described by Ettayebi and co-authors [110]. Thus, more research is needed to verify the results obtained using HIEs. Also, no replication of the three strains belonging to GIV was observed in the HIE model [113], warranting further optimization. Later, Ettayebi et al. (2021) [111] showed the successful replication of an additional five strains in HIE belonging to the GII genogroup, namely GII.6, GII.7., GII.8, GII.12, and GII.13. However, they were not able to infect HIE with other GI genotypes. The infectivity of norovirus belonging to the GII genogroup present in vomit samples has also been confirmed using the HIE model [128].


viruses-16-00776-t001_Table 1Table 1Human intestinal enteroid model for the replication of human norovirus and the detection of human norovirus present in food and water samples.StudyGenogroupStrainReferencesTesting of different HuNoV strains for replicationGIGI.1[P1][110,111,113,118,128]GIIGII.1GII.1[P41]GII.2[P2]GII.2[P16]GII.3[P12]GII.3[P21]GII.4 Yerseke[P4]GII.4 Den Haag[P4]GII.4 New Orleans[P4]GII.4 Sydney[P31]GII.4 Sydney [P16]GII.4[P16]GII.6[P7]GII.7[P7]GII.8.[P8]GII.12GII.13GII.14[P7]GII.17[P38]GII.17[P13]GII.17[P31]Persistence in seawaterGIIGII.4GII.3[114]Evaluation of thermal inactivationGIIGII.4[P16][115]Persistence in water microcosmsGIIGII.4 Sydney[P31][117]Persistence in surface waterGIIGII.4 Sydney[P31][129]Quantification in lettuce, frozen raspberries, and frozen strawberriesGIIGII.4[P16]GII.6[P7][118]


### 5.1. Advances in HIE-Culturing Techniques

As HIEs do not support the replication of all HuNoV strains, there have been efforts to optimize various factors such as culture medium, stool characteristics of the sample, HIE age, and so on. The original proliferation (BCMp) and differentiation (BCMd) medium described by [110] to grow HIEs that were later used to infect with HuNoV were compared with the commercially available medium IntestiCult™, a human organoid growth medium from Stem Cell Technologies. The results showed a significantly higher replication of HuNoV strains in the IntestiCult™medium compared to the BCM medium [111]. However, as the composition of the IntestiCult™ medium is proprietary, it is impossible to determine what factors are helping to enhance the replication of HuNoV in the presence of the IntestiCult™ medium compared to the BCM medium.

Inoculum concentration is another important factor to consider when using HIEs to study HuNoV [110,111,113,116]. The minimum HuNoV GII.4 Sydney dose required to measure growth in HIEs is reported to be approximately 10^3^ genome copies per well of a 96-well plate [110,113,116]. However, this concentration needs to be tested for additional HuNoV strains, as GII.4 has been reported to be the most efficient strain for replication in HIE models. Therefore, the minimum dose required for other strains may be considerably higher compared to GII.4.

### 5.2. HIEs for Detecting Human Norovirus Present in Food and Water Matrices

With the development of the HIE model, the assessment of HuNoV infectivity within environmental samples has become possible and has been listed in Table 1. Desdouits and co-authors (2022) [114] used the HIE model to evaluate the persistence of HuNoV in seawater, which is the matrix through which shellfish accumulate HuNoV. The stability of GII.4 and GII.3 was tested in three different seawater samples for up to 35 days by spiking 120 mL of seawater with 10^5^–10^6^ viral RNA copies/mL, and the results showed the presence of GII.4 for an average of 21 days, whereas GII.3 persisted for an average of 24 days [114]. However, it must be noted that when the input genome levels were close to 10^3^, which is the minimum amount required to infect HIE, there was no replication observed in the HIE, which suggests that infectious HuNoV particles might still be present but were just not detected by the HIE model. In another study, the viability of GII.4 Sydney[P31] in surface, tap, and deionized water microcosms was evaluated by spiking 10^6^ norovirus genome copies/mL into water samples [117]. These samples were held at room temperature (18–22 °C) in darkness. Viable HuNoV was detected for up to 28 days in tap water and deionized water, with a consistent decrease each day, after which the number of copies fell below the detection limit of the HIE model. However, in surface water, viable HuNoV did not fall below the detection limit even at 28 days. The measurement of viral RNA directly from the water microcosms revealed the RNA signal to be constant for the entire duration of the study (28 days), implying that the risk estimates based on molecular methods are usually an overestimation of the risk [117]. In a similar study, GII.4 was inoculated into filter-sterilized surface water from a freshwater creek, and HuNoV infectivity was assessed using the HIE model. The results ranged between no substantial decay in norovirus to a decay rate constant of 2.2/day [129]. Additionally, it was noted that HuNoV genome segments persisted longer than infectious HuNoV. It is important to highlight that the water sample was filter-sterilized before HuNoV inoculation, omitting the potential influence of endogenous microorganisms and other particles on HuNoV decay [129]. Norovirus has been shown to remain infectious for up to 21 days in artificial estuarine water at both 4 °C and 16 °C, with infectivity declining to 3% by day 21 [130]. However, no decrease in norovirus titers was observed using ddPCR.

With respect to the food matrix, only a couple of studies have reported using HIEs to study HuNoV in food. Hayashi et al. (2022) [115] evaluated the heat inactivation of HuNoV (GII.4) in freshwater clams using HIEs and reported that treatment at 90 °C for 1 min inactivated HuNoVs that were inoculated into the clam bodies. Recently, HuNoV strains GII.4[P16] and GII.6[P7] that were seeded onto lettuce, frozen strawberry, and frozen raspberry were recovered and quantified in HIE [118]. However, it is important to note that a high viral titer (10^7^–10^8^ genome copies) was used in both these studies to facilitate identification using the HIE model, which might not always reflect real-life scenarios.

## 6. Zebrafish Model for Detecting Infectious Human Norovirus

In 2019, a larval zebrafish model to study HuNoV was described for the first time [119]. At 3 dpf, the zebrafish larvae were injected with various strains of GI and GII HuNoV obtained from human stool samples, and replication of the virus was observed in the intestine and hematopoietic tissue of the zebrafish larvae [119]. The zebrafish model was able to support the replication of GII.3 without the addition of bile and a GI.7 strain not reported to infect HIE [113,119]. Also, the successful passaging of the GII.4 strain was demonstrated up to the second passage [119]. These findings show that zebrafish could be a very useful model to study the pathogenesis of different strains of HuNoV (Table 2).

Researchers have also explored potential antiviral strategies of small molecules and carbohydrates against HuNoV using the zebrafish larvae model [121]. In addition, the zebrafish larvae model has been employed to investigate the role of HBGAs and the host microbiota during HuNoV infection. The findings showed that successful HuNoV infection in zebrafish larvae depends on the presence of terminal fucoses, an integral part of HBGAs [120]. This aligns with infections observed in humans and in other recently established in vitro HuNoV models, emphasizing the consistent requirement for fucose residues on intestinal cells and suggesting a shared entry mechanism for HuNoV infection in zebrafish. Additionally, this study also demonstrated that neither the zebrafish microbiota nor the presence of HBGA-expressing bacteria in the zebrafish intestine amplified HuNoV replication during the early larval stages of their development [120]. This differs from the results seen in B-cells, where the presence of HBGA-expressing enteric bacteria was required for HuNoV infection [108]. Therefore, future in-depth investigation is required to determine whether the infection mechanism in zebrafish larvae differs from that in humans.

In efforts to improve the zebrafish model for studying HuNoV, Tan and coauthors (2023) [123] reported the successful replication of HuNoV through injection into zebrafish embryos. When zebrafish larvae were injected with HuNoV, a notable increase in viral genome copies was detected at 2 days post-infection (dpi), followed by a gradual decrease in viral loads from 3 dpi. In contrast, injecting the viruses into zebrafish embryos led to significant virus replication as early as 1 dpi, persisting until 6 dpi. Furthermore, the high levels of virus replication enabled continuous passaging for up to four passages [123]. Therefore, this model could be used to study the adaptive mutation of HuNoVs due to virus passaging, which has been reported in multiple viruses such as SARS-CoV-2, Hepatitis C, and Zika virus [131,132,133]. Tan and coauthors (2023) [123] also demonstrated that UV254 treatment led to a 2–4 log reduction in HuNoV infectivity for three different HuNoV strains (GII.4[P16], GII.2[P16], GII.17[P31]). This finding indicates that HuNoVs exhibit a higher susceptibility to UV inactivation when compared to frequently employed surrogates like MNV, which showed a 1.92 log reduction [134], and TuV, which displayed a 1.08 log reduction [135]. The zebrafish model for HuNoV has emerged as a promising tool for studying HuNoV infection. Its advantages in terms of genetic tractability, optical transparency until early adulthood, and physiological relevance make it a valuable alternative to traditional models. In addition to this, zebrafish are small in size, produce a high number of offspring in a short period of time, and have low maintenance and husbandry costs compared to other small-animal models [136].

### Zebrafish Model to Detect Human Norovirus Present in Food and Water Matrices

The zebrafish larvae/embryo model, being a relatively recent development, has yet to be extensively explored for its potential for the detection of HuNoV extracted from food and water. Rather, researchers are currently investigating different methods of virus administration to zebrafish larvae for HuNoV studies. The first established zebrafish larvae model for HuNoV utilized the microinjection of 3 nL of filtered virus inoculum into the yolk of zebrafish larvae, demonstrating its effectiveness as a virus administration route [119]. Immersing 5-day post-fertilization larvae in a GII.4 virus suspension was also considered, but did not yield increased viral replication, indicating that immersion is not a viable route of infection for GII.4. However, it remains necessary to evaluate other HuNoV strains to determine the viability of the immersion method. Another potential route of infection involves using *Paramecium caudatum*, a ciliated protozoan and natural prey for zebrafish larvae, as a vehicle for a foodborne infection model. This method has been successfully employed in studying infections caused by *Escherichia coli* and *Salmonella* Typhimurium in zebrafish larvae [137,138,139]. Although there are currently no reports of using this foodborne infection model for HuNoV, a study involving two HuNoV surrogates, murine norovirus (MNV) and feline calicivirus (FCV), demonstrated their persistence within two species of free-living amoeba (*Acanthameoba castellanii* and *A. polyphaga*) for up to 8 days [140]. This suggests the possibility that HuNoV could also remain stable within *P. caudatum*, which could then be fed to zebrafish larvae to induce an infection. The culturing of *P. caudatum* free from other microorganisms is feasible [141], eliminating concerns about other microbial infections. Furthermore, if *P. caudatum* associated with HuNoV is found to induce infections in zebrafish larvae, this method could potentially be adapted for the detection of HuNoV in food and water matrices.

## 7. Challenges in Using the HIE Model and Zebrafish Model to Detect Virus Presence in Food and Water Matrices

### 7.1. Sample Matrix

The HIE and the zebrafish model for studying HuNoV are relatively new, and therefore no prior investigations into the extraction of HuNoV from food matrices, such as fresh produce, and the subsequent inoculation of either HIE or zebrafish embryos or larvae have been reported. Furthermore, most previous studies have primarily focused on the extraction of viral RNA rather than isolating intact virions [142]. Also, the sample matrix may contain many other elements such as carbohydrates, lipids, and other microorganisms that can act as inhibitors or cause contamination of the HIE models, or can be toxic when injected into zebrafish larvae without prior filtration. Currently, clarified stool samples have been used to infect both HIEs and zebrafish larvae/embryos [110,111,123]. However, this approach limits our ability to explore the impact of the gut microbiota on HuNoV infection. While some studies have successfully colonized the zebrafish larvae with some members of human gut microbiota [143,144,145], further research is warranted to co-inoculate zebrafish larvae with both human gut microbiota and HuNoV. Also, the food type will influence the recovery of HuNoV, especially at the concentrations needed for these culture models to successfully detect the virus. While it was possible to recover at least some virus from foods such as penne pasta salad and macaroni, no virus inocula could be recovered from deli sandwiches and roast beef meat [99]. Further methods need to be developed while adopting these models to assess the presence of HuNoV in different food and water matrices.

### 7.2. Virus Availability and Concentration

As stated previously, HuNoV does not have a continuous cell culture system that allows for easy propagation and study in a laboratory setting. This makes it difficult to obtain and maintain a consistent source of HuNoV. Researchers have predominantly depended on human stool samples derived from clinical cases as their primary resource for research. Also, not all strains are readily available for research. Strains belonging to the genogroup GIV have recently been reported to infect humans [146]. Additionally, another rare norovirus genotype, GIX.1, which was reported in 1990 to have caused a large outbreak of acute gastroenteritis [26], is being detected sporadically [147]. However, due to their limited circulation, obtaining a sufficient quantity of these less-common strains for research has proven to be arduous. Additionally, not all fecal samples obtained from patients contain HuNoV at a high enough titer. Given that HIE and zebrafish models necessitate an initial higher titer for effective infection [111,119,123], having an initially lower concentration could impede their utility in research studies. Consequently, researchers may need to employ virus concentration methods before utilizing these models for their investigations.

### 7.3. Strain Specificity

As previously discussed, it is worth noting that not all HuNoV strains exhibit the capability to infect HIEs and zebrafish larvae. Among those strains that have shown infectivity in HIEs, the majority belong to the GII genogroup [111]. Within the GI genogroup, only GI.1 has demonstrated the potential to infect HIEs, although these findings have not been independently verified by another research team [111,113]. GIV virus strains have proven unsuccessful in replicating within HIEs [113]. Concerning the zebrafish larvae model, once again, a predominant trend has been observed, with most strains that were tested and found to successfully infect zebrafish larvae or embryos belonging to the GII genogroup [121,123]. To date, no research has explored the infectivity of GIV strains in zebrafish larvae, and only GI.7 within the GI genogroup has been shown to infect this model [119]. Hence, further research is needed to enhance the effectiveness of these models and gain a comprehensive understanding of the factors influencing HuNoV pathogenesis. This will enable both the HIE model and the zebrafish larvae/embryo model to emerge as robust platforms for studying the diverse spectrum of HuNoV strains.

### 7.4. Time and Cost Constraints

The HIE model represents a labor-intensive and costly approach. The HIE medium is both intricate and expensive due to the necessity of various growth factors, including Wnt-3A, R-Spondin, and Noggin, which require the cultivation of three distinct cell lines [110]. Although there is now the availability of a single cell line capable of producing all three growth factors, along with commercial media options, these additions further escalate the already substantial expenses associated with the HIE model [111]. The reagents required for testing a single sample of HuNoV in HIE culture incur an approximate cost of USD 36, and the entire process, from initiation to quantification, demands a minimum duration of three weeks [116]. In contrast, the zebrafish model emerges as a more cost-effective and less labor-intensive alternative. Additionally, zebrafish possess the advantages of small size, high reproductive rates within a short timeframe, and comparatively modest maintenance and husbandry costs [136].

### 7.5. Ethical Issues Concerning the Zebrafish Model

Zebrafish have a corresponding orthologue for almost 70% of human genes and the development and physiology of the GI tract bears similarities to humans [148,149]. Hence, the zebrafish larvae model offers a promising avenue to study HuNoV mirroring for several key aspects of human physiology while also being more ethically aligned with the replacement principles of the 3Rs—Replacement, Reduction and Refinement—originally introduced by Russel and Burch in 1959 [150]. Its genetic proximity to humans positions the zebrafish model as a suitable substitute for higher-order animals, reducing the need to use them in HuNoV research. As zebrafish produce a high number of offspring in a short period of time, the ability to conduct larger-scale studies with fewer animals not only minimizes the overall use of animals but also enhances the statistical power and robustness of experimental results, reducing the need for redundant experimentation. Early-stage zebrafish animals (3–7 dpf) do not feel pain or distress and the zebrafish larvae and embryo model involves procedures such as microinjection to be carried out before 3 dpf [119,123]. This minimizes the distress that the animals are put through during the experimentation process and hence comes in line with the final R, which is refinement.

## 8. Conclusions and Future Research

The HIE and zebrafish larvae/embryo models offer valuable insights into HuNoV pathogenesis and the assessment of treatment strategies for HuNoV infections. Nevertheless, several challenges arise when applying these models to detect viruses in food and water matrices. Both models demand high viral titers as an initial inoculum, necessitating further optimization to analyze food and water samples that typically contain limited viable virus quantities, resulting in low recovery rates. Additionally, the extraction and concentration methods for HuNoV from various food matrices and water samples must be tailored to meet the models’ requirements, considering potential contaminants that may be toxic to both systems. Furthermore, it is crucial to explore viruses within genogroups GI (except GI.1 for HIE and GI.7 for zebrafish larvae) and GIV that have not yet been studied or have demonstrated infectivity using HIE and zebrafish models. This exploration is essential for the comprehensive detection of diverse HuNoV strains in various matrices and a more profound understanding of the pathogenesis of each strain. Such insights are instrumental in developing strategies to mitigate HuNoV infections. Finally, both HIE and zebrafish larvae models rely on the detection of viral RNA post-infection, which may not always provide an accurate measure of the infectious viral RNA content within them.

## Figures and Tables

**Table 2 viruses-16-00776-t002:** Zebrafish larvae and embryo model for the replication of human norovirus.

Study	Genogroup	Strain	References
Testing of different HuNoV strains for replication in zebrafish larvae	GI	GI.7[P7]	[119,121]
	GII	GII.2[P16] GII.3[P16] GII.4[P4] GII.4[P16] GII.6[P7] GII.17[P31]	
Testing of different HuNoV strains for replication in zebrafish embryo	GII	GII.2[P16] GII.4[P16] GII.17[P31]	[123]

HuNoV—human norovirus.

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
