# Peer review of "Improving the Detection and Understanding of Infectious Human Norovirus in Food and Water Matrices: A Review of Methods and Emerging Models"

_viruses, 2024, doi:10.3390/v16050776_

Round 1
Reviewer 1 Report
Comments and Suggestions for Authors
1. The title is not enough accurate to summarize the contents of this review. Actually, the review did not largely introduce detection methods or challenges to some extent. Furthermore, the subtitle is also not suitable because of some inclusions between the subtitles, such as subtitle 4 and 5, 6, etc. The author should refine these titles.
2. “Recovery” in Line 169 is not accurate. In this manuscript the recovery is not associated with the virus infectivity, which only includes the virus enrichment and extract.
3. The author could show more your own opinions in this review.
Comments on the Quality of English Language1. The genome copies should be 10 to the power of 3, 4 or 5 in Line 375, 376, 421,424, etc.
2. The symbol of temperature is also incorrect in Line 429, 448, etc.
3. The genogroup I of norovirus is named as GI or G1 in this manuscript, which should be consistent in this manuscript.
4. In Line 386, 388 and 401, the author should clarify who or which research team, not be referenced by 111.
5. The font size of reference 70 is different from others.
Besides, there are many similar problems in this manuscript. The author should proof the manuscript carefully.
Author Response
Reviewer 1:
Comments and Suggestions for Authors:
- The title is not enough accurate to summarize the contents of this review. Actually, the review did not largely introduce detection methods or challenges to some extent. Furthermore, the subtitle is also not suitable because of some inclusions between the subtitles, such as subtitle 4 and 5, 6, etc. The author should refine these titles.
Response: The authors appreciate the reviewer’s feedback. The title has been revised to “Improving detection and understanding of infectious human norovirus in food and water matrices: A review of methods and emerging models” to better encapsulate the content of the review paper. Additionally, subtitles 5 and 6 have been revised to include “for detecting infectious human norovirus” after mentioning human intestinal enteroids model and zebrafish model, respectively, for improved clarity. We believe that subtitle 4 “Cell culture and animal models available for detecting infectious human norovirus” is appropriate as it succinctly summarizes all currently available models supporting replication of human norovirus.
- “Recovery” in Line 169 is not accurate. In this manuscript, the recovery is not associated with the virus infectivity, which only includes the virus enrichment and extract.
Response: We thank the reviewer for the comment. The subtitle has been revised to “Extraction and concentration methods”.
- The author could show more your own opinions in this review.
Response: We appreciate the reviewer’s point of view. However, this is intended to be an unbiased review of the literature and current evidence, not a commentary or op-ed. Thus, we respectively decline to add our opinion.
Comments on the Quality of English Language
- The genome copies should be 10 to the power of 3, 4 or 5 in Line 375, 376, 421,424, etc.
Response: We thank the reviewer for bringing these formatting errors to our attention. They have been rectified throughout the manuscript.
- The symbol of temperature is also incorrect in Line 429, 448, etc.
Response: We apologize for the oversight in formatting the temperature symbol correctly. It has now been corrected throughout the manuscript.
- The genogroup I of norovirus is named as GI or G1 in this manuscript, which should be consistent in this manuscript.
Response: Genogroup I of norovirus has been abbreviated as GI throughout the manuscript.
- In Line 386, 388 and 401, the author should clarify who or which research team, not be referenced by 111.
Response: We appreciate the reviewer’s comment. The sentence has been revised to explicitly mention the research team.
- The font size of reference 70 is different from others.
Response: We apologize for the oversight. The reference font has been adjusted to align with the manuscript’s font.
- Besides, there are many similar problems in this manuscript. The author should proof the manuscript carefully.
Response: The manuscript has been carefully proofread once more, and all formatting and grammatical errors have been addressed and corrected.
Reviewer 2 Report
Comments and Suggestions for Authors
This is a very well written review article on the current state of challenges associated with norovirus detection in food and environmental samples. The authors focused mainly on the various methods for norovirus detection from various samples, their advantages and shortcomings along with the models that are currently available or in development for norovirus replication.
The initial sections (Introduction and transmission routes), have focused on the basics of norovirus biology and the various routes through which human norovirus could be transmitted, while discussing the importance of having a model for norovirus replication and how it could help us better address some of the unknowns of norovirus biology like selective binding of GI over GII strains to oysters. The authors focus on these rather significant findings from various research groups which help us appreciate the genetic diversity of norovirus genogroups.
The references cited in the article are appropriate to the review article and
It would be nice to see more information on the significance of sample concentration prior to detection as this is a pivotal step in the detection of human norovirus are often overlooked. There are a multitude of sample concentration methods that have yielded better results with the existing detection methods which would otherwise underestimate the virus load due to inhibitory substances found in the matrices that are being tested. The authors should have dedicated a small section on the various sample pre treatment methods to signify its importance in detection.
Author Response
Comments and Suggestions for Authors:
This is a very well written review article on the current state of challenges associated with norovirus detection in food and environmental samples. The authors focused mainly on the various methods for norovirus detection from various samples, their advantages and shortcomings along with the models that are currently available or in development for norovirus replication.
The initial sections (Introduction and transmission routes), have focused on the basics of norovirus biology and the various routes through which human norovirus could be transmitted, while discussing the importance of having a model for norovirus replication and how it could help us better address some of the unknowns of norovirus biology like selective binding of GI over GII strains to oysters. The authors focus on these rather significant findings from various research groups which help us appreciate the genetic diversity of norovirus genogroups.
The references cited in the article are appropriate to the review article and
It would be nice to see more information on the significance of sample concentration prior to detection as this is a pivotal step in the detection of human norovirus are often overlooked. There are a multitude of sample concentration methods that have yielded better results with the existing detection methods which would otherwise underestimate the virus load due to inhibitory substances found in the matrices that are being tested. The authors should have dedicated a small section on the various sample pre treatment methods to signify its importance in detection.
Response: We appreciate the reviewer for emphasizing the significance of the concentration step preceding the detection of norovirus in food and water samples. In section 3, "Detection of human norovirus in food and water," we have thoroughly examined several concentration techniques described by different researchers, including PEG precipitation, adsorption-elution on an electropositive microporous filter, ultrafiltration, proteinase K treatment, organic flocculation, immunomagnetic separation, and ultracentrifugation for concentrating norovirus present in water and food matrices.